# Quantum Embedding of Knowledge for Reasoning

**Dinesh Garg**[1*], **Shajith Ikbal**[1*], **Santosh K. Srivastava**[1], **Harit Vishwakarma**[2†],
**Hima Karanam**[1], **L Venkata Subramaniam**[1]
[1]IBM Research AI, India
[2]Dept. of Computer Sciences, University of Wisconsin-Madison, USA
garg.dinesh, shajmoha, sasriva5@in.ibm.com, hvishwakarma@cs.wisc.edu, hkaranam, lvsubram@in.ibm.com

## Abstract

Statistical Relational Learning (SRL) methods are the most widely used techniques to generate distributional representations of the symbolic Knowledge Bases (KBs). These methods embed any given KB into a vector space by exploiting statistical similarities among its entities and predicates but without any guarantee of preserving the underlying logical structure of the KB. This, in turn, results in poor performance of logical reasoning tasks that are solved using such distributional representations. We present a novel approach called *Embed2Reason* (*E2R*) that embeds a symbolic KB into a vector space in a **logical structure preserving manner**. This approach is inspired by the theory of Quantum Logic. Such an embedding allows answering membership based complex logical reasoning queries with impressive accuracy improvements over popular SRL baselines.

## 1 Introduction

We consider the problem of embedding a given symbolic Knowledge Base (KB) into a vector space that preserves the **logical structure**. Such embeddings are popularly known as *distributional representation* (e.g., Word2Vec [1] and GLOVE [2]) and are aimed to be leveraged by several non-symbolic (e.g. neural and vector) methods to accomplish various tasks (e.g. KB completion/ link prediction and logical reasoning under noisy/incomplete KB) on which symbolic methods struggle.

Figure 1 (best viewed in color) depicts a toy example of the kind of symbolic KBs that we wish to embed into a vector space. The red-colored oval-shaped nodes of the left tree denote a hierarchy (aka ontology) of the unary predicates (aka concepts), and the blue-colored rectangle-shaped nodes denote the memberships of entities to various concepts. Similarly, the right tree denotes an ontology of binary predicated (aka relations) and the ordered pair of entities as their members. In the right tree, having (Bob, Alice) as a child node of the relation Father_of means Bob is Father_of Alice. In such ontologies, a predicate node logically implies (aka entails) any of its ancestral predicate node. For example, cardiologist ⇒ physician ⇒ doctor, and similarly, Mother_of ⇒ Parent_of ⇒ Blood_relation. In the framework of mathematical logic, such a logical structure can be expressed via a subset of Description Logic (DL) statements. In the DL parlance, logical structure among predicates (red oval nodes) is commonly known as T-box, whereas logical structure connecting entities (or entity tuples) to the predicates is commonly known as A-box [3].

As far as vector embeddings of symbolic KBs is concerned, last couple of years have witnessed a surge in the newer methods with an end goal of KB completion or logical inferencing to answer reasoning queries [4, 5, 6, 7, 8, 9, 10, 11, 12]. While details being different, a high level idea behind all these methods is as follows – Ingest a given symbolic KB and embed it into a vector space by using a neural network; followed by conversion of logical queries into score based algebraic operations

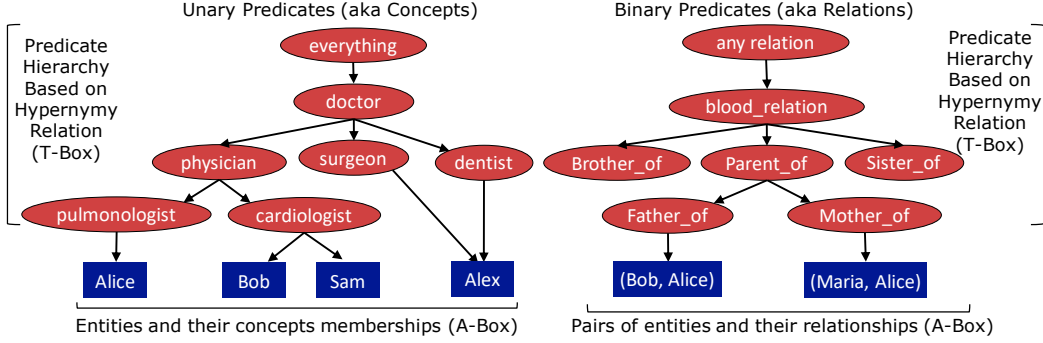

**Figure 1:** A toy example of knowledge base. The left (right) figure depicts unary (binary) predicates ontology.

over the vector space. Unlike pure symbolic logic frameworks [13, 14, 3], these neural methods are robust against input noise but at the same time, they suffer from a major drawback, namely *no guarantees that embeddings maintain the sanctity of the logical structure present in the input A-Box and T-Box*. Therefore, unlike pure symbolic reasoners, a machine that uses such embeddings for complex reasoning tasks struggle with the accuracy. The embeddings resulted by some of these neural methods [4, 5, 6, 7, 8] capture, at best, the logical structure of just A-box but not T-box. Some other methods [9, 10, 11], which provision to maintain T-box structure, suffer from the fact that resulting logical structure in the embedding is not sound enough by the design (unlike ours which is based on Quantum Logic) and hence they often fail to meet high accuracy for complex reasoning tasks.

In this paper, we propose a novel approach, called *Embed2Reason* (*E2R*) that can embed a given symbolic KB (such as the one shown in Figure 1) into a finite dimensional vector space. E2R is inspired from the theory of Quantum Logic (QL) which guides us constraining the embedding in such a way that *the set of all logical propositions in the input A-Box and T-Box become isomorphic to a lattice structure over a set of subspaces of the vector space*. Such an embedding satisfies the axioms of the Quantum Logic [15, 16] and allows one to perform logical operations (e.g. negation, conjunction, disjunction, and implication) directly over the vectors in a manner similar to the Boolean Logic except that distributive law does not hold true [17]. We call such an embedding as *Quantum Embedding*. We formulate an unconstrained optimization program that captures all such quantum logical (as well as regularity) constraints. We solve this program via Stochastic Gradient Descent (SGD) technique and the resulting embedding maintains the input logical structure with a good accuracy. Next, we show that these quantum embeddings can solve complex deductive as well as predictive reasoning tasks (formally defined later) with a much superior performance as compared to other kinds of embeddings. Specifically, quantum embeddings show better performance on both link prediction as well as reasoning tasks. In our experiments, we found that on FB15K dataset, quantum embeddings obtained via the proposed E2R method exhibit $57\%$ improvement on MRR and $95\%$ improvement on HITS@1. Further, for LUBM1U dataset, we found E2R exhibiting $76\%$ improvement on MRR and $139\%$ improvement on HITS@1 relative to the closest competitor.

## 2 Preliminaries

**Description Logic (DL) Syntaxes** Description Logics (DLs) are a family of logics that are fragments of First Order Logic (FOL) [3, 18]. In this paper, we confine to the simplest form of DL, namely *Attributive Language with Complements ($\mathcal{ALC}$)*. A triple $(\mathcal{N}_O, \mathcal{N}_C, \mathcal{N}_R)$ is called as the *signature of the DL* where, $\mathcal{N}_O = \{O_1, O_2, \ldots, O_{|\mathcal{N}_O|}\}$ denotes a finite set of entities (aka objects or elements) – blue rectangular nodes in the left tree of Figure 1. $\mathcal{N}_C = \{C_1, C_2, \ldots, C_{|\mathcal{N}_C|}\}$ denotes a finite set of unary predicates (aka (atomic) concepts, classes, or types) – red oval nodes in the left tree of Figure 1. $\mathcal{N}_R = \{R_1, R_2, \ldots, R_{|\mathcal{N}_R|}\}$ denotes a finite set of binary predicates (aka (atomic) relations or roles) – red oval nodes in the right tree of Figure 1. The complex logical statements in DL are formed by applying one or more of the following constructors on one or more of the atomic concepts: *negation* ($\neg$) of a concept, *equality* ($=$), *intersection* ($\sqcap$), *union* ($\sqcup$), and *logical inclusion* ($\sqsubseteq$) between two concepts. Also, there are two special concepts called $\top$ (every concept) and $\bot$ (empty concept). Finally, the operators *universal restriction ($\forall R.C$)* and *existential restriction ($\exists R.C$)* correspond to the concepts

whose members are given by the sets $\{O \in \mathcal{N}_O \mid \forall O'$ having $R(O, O')$ we must have $O' \in C\}$ and $\{O \in \mathcal{N}_O \mid \exists\ O'$ having $R(O, O') \wedge (O' \in C)\}$, respectively.

**Formal Concept Analysis (FCA) and Distributional Representation**   FCA [19] is a branch of lattice theory [20] focusing on special type of lattices that are obtained from a set of *entities*, a set of *attributes*, and a binary relation called *context* that determines whether an entity has an attribute. Although not directly related, FCA has lots of similarities with DL and can be exploited to get a naive and partial solution of the central question in this paper. FCA theory allows embedding entities of any DL into a binary space, say $\{0, 1\}^d$. Such an embedding can also be used as a device to perform Boolean logical operations on unary predicates. Supplementary material comprises further details.

## 3   Quantum Embeddings

Although, FCA based embedding partially preserves the logical structure of DL, it has several limitations - (i) FCA makes closed world assumption – *all the missing membership assertions are treated as non-members in FCA*. In real world, this assumption is rarely true. In fact, we don't even know how many other atomic concepts/relations are missing in the given KB. (ii) In many real world setup, an entity may have soft/fuzzy membership to certain concepts - for example, *an apple may have each of red and green color but to a varying extent*. (iii) Lastly, FCA embeds only unary predicates but we want to embed both unary and binary predicates. These limitations of FCA based embeddings motivated us to explore the use of Quantum Logic. [3]

Originally, QL [22, 17] was proposed by [15] to model the quantum behavior of subatomic particles. We, however, leverage it here to embed A-box and T-box of a DL while preserving the logical structure. In QL, every predicate (unary or binary as well as atomic or compound) is represented by a linear subspace of a complex vector space $\Sigma$, where $\Sigma = \mathbb{C}^d$ for some integer $d$. [4] All the entities and entity pairs are denoted by (complex) vectors and they lie in the predicate subspaces to which they belong (see Figure 2 for a graphical illustration of the idea). The axes of such an embedding space represent latent semantic attributes of the entities and entity pairs. The set of all linear subspaces of $\Sigma$, known as *Grassmannian*, can be put up in correspondence with the set of all possible predicated - infinitely many of them are not even supplied as part of typical input KB. The subspaces of $\Sigma$ naturally have a partial order relation induced over them by set theoretic inclusion operation. That is, for any two subspaces $S_i, S_j \subseteq \Sigma$, we say $S_i \sqsubseteq S_j$ iff $S_i \subseteq S_j$ or equivalently, $S_i$ is a subspace of $S_j$ also. Because origin is a zero dimensional subspace and is common to any subspace, by letting $\bot = \{0\}$ and $\top = \Sigma$, the resulting partial order over the subspaces becomes bounded lattice of $\infty$ size. The quantum logical operations on this lattice are defined by its inventors [15] as follows.

- $S_i \wedge S_j = S_i \sqcap S_j := S_i \cap S_j$. That is, logical `AND` or `intersection` between two subspaces is equal to the largest common subspace.

- $S_i \vee S_j = S_i \sqcup S_j := S_i + S_j$. Here $+$ sign means the *vector sum* and not the set theoretic union. That is, logical `OR` or `union` between two subspaces is the smallest subspace encompassing both these subspaces. This is where QL differs from the Boolean Logic. This axiom also results in non-distributive nature of QL. That is, the distributive law $S_i \wedge (S_j \vee S_k) = (S_i \wedge S_j) \vee (S_i \wedge S_k)$ holds no more true in QL. However, we have proven (refer supplementary material) that by restricting each predicate subspace to be parallel to the axes, we can avoid this limitation of the QL.

- $\neg S := S^\perp$, where $S^\perp$ is known as *orthogonal complement* of the subspace $S$. Every vector in $S^\perp$ is perpendicular to every vector in $S$ and also $S + S^\perp = \Sigma$. This means, logical `NOT` or `negation` of any subspace is given by set of all those vectors that are normal to this subspace.

Figure 2 depicts an illustrative example of Quantum Logic based embedding of the concept ontology into, say a $\Sigma = \mathbb{C}^3$ space. The embedding of relation hierarchy would also be similar except that an ordered entity pair such as (Bob, Alice) would be mapped to the vector $(\boldsymbol{V}_{Bob} + \iota \boldsymbol{V}_{Alice})$, where $\boldsymbol{V}_{Bob} + \iota \boldsymbol{0}, \boldsymbol{V}_{Alice} + \iota \boldsymbol{0}$ are the embedding vectors for entities Bob, Alice, respectively. In other words, the imaginary component of the entities would be zero but for a pair of entities it would be non-zero. The relations would be mapped to different linear subspaces. In next section, we

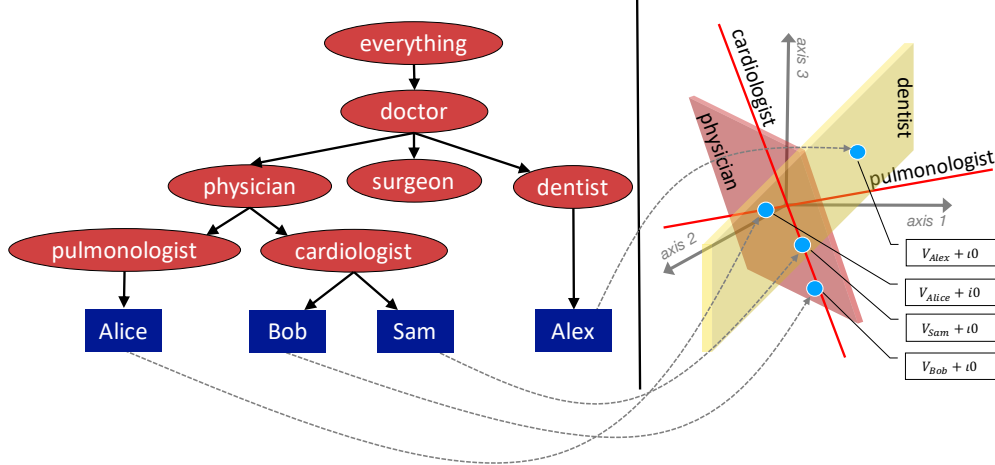

**Figure 2:** An illustrative example of Quantum embedding

present E2R learning scheme which automatically maps these predicates and (pair of) entities to the subspaces and vectors, respectively, in the space $\Sigma = \mathbb{C}^d$ for some integer $d$.

## 4 Embed2Reason

E2R writes down a quantum logical constrain for every logical structure present in A-Box/T-Box so as to ensure it preservation during embedding. The overall problem, thus, becomes a constraint satisfaction problem which we convert into an unconstrained optimization problem via loss functions.

In E2R, we assume the underlying field of the embedding space $\mathbb{C}^d$ is $\mathbb{R}$. Under this assumption - (i) The vector space $\Sigma = \mathbb{C}^d$ becomes isomorphic to the Euclidean space $\mathbb{R}^{2d}$. That means, there exist a bijection map $T : \mathbb{C}^d \mapsto \mathbb{R}^{2d}$ which preserves vector addition and scalar multiplication [23]. (ii) The subspace corresponding to any single axis of $\mathbb{C}^d$ can be viewed as span of two subspaces – real part and imaginary part. This, in turn, implies that the set of vectors $\{x_R + \iota 0\}$ forms a valid subspace of $\mathbb{C}^d$. (iii) Any axis-parallel subspace of $\mathbb{C}^d$ can be mapped to a unique axis-parallel subspace of $\mathbb{R}^{2d}$ and vice-a-versa. Due to these properties, we can simulate $\mathbb{C}^d$ via $\mathbb{R}^{2d}$ on a digital computer. Specifically, any axis parallel subspace $S$ of $\mathbb{C}^d$ can be simulated in the space $\mathbb{R}^{2d}$ by means of defining a $2d$-dimensional 0/1 indicator vector $y$ having the following property – The coordinates $y[i]$ and $y[i + d]$ would be 1 (or 0) depending on whether real and imaginary part, respectively, of the $i^{th}$-axis $(1 \leq i \leq d)$ of $\mathbb{C}^d$ is part (or not part) of the subspace $S$. To this end, we make a simplifying assumption that E2R always outputs an embedding where subspace corresponding to any predicate is parallel to the axes of the space $\mathbb{C}^d$. We call such an embedding as *axis-parallel embedding*. The axis-parallel embedding assumption alleviates the limitation of the QL regarding distributive law (refer supplementary material). Lastly, because the underlying field is $\mathbb{R}$, we use the following valid inner product formula [23]: $\langle x, y \rangle = [(y^{*\top}x)^*(y^{*\top}x)]^{1/2}$ for $x, y \in \mathbb{C}^d$, where $*$ denotes the complex conjugate operation. This inner product is essentially the length of the complex number obtained by taking the standard inner product between two complex vectors [23]. With this setup, we now describe our E2R technique.

**Entities, Pair of Entities, and Predicates** E2R maps individual entities $O_i, O_j \in \mathcal{N}_O$ to the unit length vectors (under $\ell_2$ norm) of the form $x_i = \begin{pmatrix} x_{iR} \\ 0 \end{pmatrix}$ and $x_j = \begin{pmatrix} x_{jR} \\ 0 \end{pmatrix}$, respectively, and the ordered pair $(O_i, O_j)$ to the vector $x_{ij} = \begin{pmatrix} x_{iR} \\ x_{jR} \end{pmatrix}$. This can automatically be ensured by provisioning following loss terms, for each entity $O_i$, in the optimization program of E2R.

$$L_{O_i} = \left\| x_i \odot \begin{pmatrix} 0_d \\ 1_d \end{pmatrix} \right\|^2 \tag{1}$$

where, $0_d$ and $1_d$ are $d$-dimensional vectors of all $0s$ and $1s$, and $\odot$ denotes element wise multiplication. Observe that having $L_{O_i} = 0$ is both necessary and sufficient condition for the $x_i$ to have the form $\begin{pmatrix} x_{iR} \\ 0 \end{pmatrix}$. Therefore, including $L_{O_i}$ as a loss term will push its value to go towards zero.

Recall, QL theory requires entities (and pair of entities) to lie in the subspaces of concepts (and relations) to which they belong. Therefore, above representation conventions would immediately

entail that the subspace corresponding to any unary predicate, say $C_i$, must be given by an indicator vector $\boldsymbol{y}_i$ having property that $\boldsymbol{y}_i[t] = 0 \ \forall (d+1) \leq t \leq 2d$. Similarly, the subspace corresponding to any binary predicate, say $R_i$, must be given by the indicator vector $\boldsymbol{z}_i$ having property that $\exists \ (1 \leq t_1 \leq d)$ and $(d+1 \leq t_2 \leq 2d)$ such that $\boldsymbol{z}_i[t_1] = \boldsymbol{z}_i[t_2] = 1$. These constraints can be implemented via following loss terms.

$$L_{C_i} = \left\| \boldsymbol{y}_i \odot \left( \begin{smallmatrix} \mathbf{0}_d \\ \mathbf{1}_d \end{smallmatrix} \right) \right\|^2 ; \ L_{R_i} = \left( \min \left\{ 0, \left( \boldsymbol{z}_i^\top \left( \begin{smallmatrix} \mathbf{0}_d \\ \mathbf{1}_d \end{smallmatrix} \right) - 1 \right) \right\} \right)^2 + \left( \min \left\{ 0, \left( \boldsymbol{z}_i^\top \left( \begin{smallmatrix} \mathbf{1}_d \\ \mathbf{0}_d \end{smallmatrix} \right) - 1 \right) \right\} \right)^2 \quad (2)$$

It is noteworthy that above constraints would not make sense unless we ensure indicator vectors $\boldsymbol{y}_i$ and $\boldsymbol{z}_i$ are binary vectors. Such combinatorial constraints can be approximately enforced by provisioning the following two losses.

$$L_{\boldsymbol{y}_i} = \left\| \boldsymbol{y}_i \odot \overline{\boldsymbol{y}}_i \right\|^2 ; \ L_{\boldsymbol{z}_i} = \left\| \boldsymbol{z}_i \odot \overline{\boldsymbol{z}}_i \right\|^2 \tag{3}$$

where, $\overline{\boldsymbol{y}_i}$ (or $\overline{\boldsymbol{z}_i}$) is the bit flipped version of $\boldsymbol{y}_i$ (or $\boldsymbol{z}_i$) given by $\overline{\boldsymbol{y}_i} = \mathbf{1}_{2d} - \boldsymbol{y}_i$ (or $\overline{\boldsymbol{z}_i} = \mathbf{1}_{2d} - \boldsymbol{z}_i$). The dimensions of the subspaces for $C_i$ and $R_i$, and the set of corresponding basis vectors (given by indicator vectors $\boldsymbol{y}_i$ and $\boldsymbol{z}_i$) are learnable parameters and are learned automatically.

**Membership** For any given $O_i \in C_j$, or $R_i(O_p, O_q)$, E2R aspires to ensure that vectors $\boldsymbol{x}_i$ or $\boldsymbol{x}_{pq}$ lie in the subspaces corresponding to $C_i$ and $R_i$, respectively. We felt a natural way to model such constraints would be via defining residual length of the projection as a loss metric. The same idea is also endorsed by the Quantum Logic where projection length is indeed used as a probability of such membership assertion. This can be ensured by means of the following loss terms which essentially project the vectors $\boldsymbol{x}_i$ and $\boldsymbol{x}_{pq}$ into subspaces $\boldsymbol{y}_j$ and $\boldsymbol{z}_i$ in an orthogonal manner and enforce zero residual components of these projections. These loss terms also take care of the unit lengths for entity vectors. Here, $\mathbf{0}_d$ represents a $d \times d$ matrix of all $0s$.

$$L_{(O_i \in C_j)} = \left\| \overline{\boldsymbol{y}_j} \odot \boldsymbol{x}_i \right\|^2 + \left( \mathbf{1}_{2d}^\top \left( \left( \left( \begin{smallmatrix} \mathbf{0}_d & \mathbf{I}_d \\ \mathbf{I}_d & \mathbf{0}_d \end{smallmatrix} \right) \overline{\boldsymbol{y}_j} \right) \odot \boldsymbol{x}_i \right) \right)^2 + \left( 1 - \boldsymbol{x}_i^\top \boldsymbol{x}_i \right)^2 ; \quad L_{R_i(O_p, O_q)} =$$

$$\left\| \overline{\boldsymbol{z}_i} \odot \boldsymbol{x}_{pq} \right\|^2 + \left( \mathbf{1}_{2d}^\top \left( \left( \left( \begin{smallmatrix} \mathbf{0}_d & \mathbf{I}_d \\ \mathbf{I}_d & \mathbf{0}_d \end{smallmatrix} \right) \overline{\boldsymbol{z}_i} \right) \odot \boldsymbol{x}_{pq} \right) \right)^2 + \left\| \left( \begin{smallmatrix} \mathbf{1}_d \\ \mathbf{0}_d \end{smallmatrix} \right) \odot \boldsymbol{x}_{pq} - \boldsymbol{x}_p \right\|^2 + \left\| \left( \begin{smallmatrix} \mathbf{0}_d \\ \mathbf{1}_d \end{smallmatrix} \right) \odot \boldsymbol{x}_{pq} - \boldsymbol{x}_q \right\|^2 (4)$$

The last term in the expression of $L_{(O_i \in C_j)}$ tries to enforce unit length constraint for each entity vector. The last two terms in the expression of $L_{R_i(O_p, O_q)}$ try to enforce the structure of $\boldsymbol{x}_{pq}$ to be equal to $\left( \begin{smallmatrix} \boldsymbol{x}_{pR} \\ \boldsymbol{x}_{qR} \end{smallmatrix} \right)$ (as defined earlier). The second term in each of the above losses represents squared length of complex part of the projected vector. The second term would be be zero for $L_{(O_i \in C_j)}$ due to the assumptions about structures of $\boldsymbol{x}_i$ and $\boldsymbol{y}_j$.

**Logical Inclusion** As per the axioms of QL, for any given $C_i \sqsubseteq C_j$ (or $R_i \sqsubseteq R_j$), the subspace corresponding to $C_i$ (or $R_i$) must be a subset of the subspace corresponding to $C_j$ (or $R_j$). Under axis-parallel assumption, such a constraint can be implemented via the following loss term.

$$L_{C_i \sqsubseteq C_j} = \left\| \boldsymbol{y}_i \odot \overline{\boldsymbol{y}_j} \right\|^2 ; \ L_{R_i \sqsubseteq R_j} = \left\| \boldsymbol{z}_i \odot \overline{\boldsymbol{z}_j} \right\|^2 \tag{5}$$

where $\boldsymbol{y}_i, \boldsymbol{y}_j, \boldsymbol{z}_i, \boldsymbol{z}_j$ are the indicator vectors for the subspaces corresponding to $C_i, C_j, R_i, R_j$, respectively. The rational behind above loss term is as follows. If $C_i \sqsubseteq C_j$ then a *necessary* condition would be to have 1 in the vector $\boldsymbol{y}_j$ at all those positions wherever there is 1 in the vector $\boldsymbol{y}_i$ and that would mean we must have $\boldsymbol{y}_i^\top \odot \left( \mathbf{1} - \boldsymbol{y}_j \right) = 0$. Similarly, we can argue that $\boldsymbol{y}_i^\top \odot \left( \mathbf{1} - \boldsymbol{y}_j \right) = 0$ is also a *sufficiency* condition for $C_i \sqsubseteq C_j$. The argument for $R_i \sqsubseteq R_j$ is similar.

**Logical Conjunction (Disjunction)** For any given logical conjunction (disjunction) of the form $C_i = C_j \sqcap C_k$ ($C_i = C_j \sqcup C_k$) and similarly for $R_i, R_j, R_k$, we can provision the following loss terms *whose value being zero provides necessary and sufficiency conditions to satisfy the QL axiom.* Here, $\max(\cdot)$ denotes the component wise max operation. Extension to multiple concepts (or predicates) such as $C_i = C_j \sqcap C_k \sqcap C_\ell$ ($C_i = C_j \sqcup C_k \sqcup C_\ell$) is straightforward.

$$L_{(C_i = C_j \sqcap C_k)} = \left\| \boldsymbol{y}_i - \left( \boldsymbol{y}_j \odot \boldsymbol{y}_k \right) \right\|^2 ; \ L_{(R_i = R_j \sqcap R_k)} = \left\| \boldsymbol{z}_i - \left( \boldsymbol{z}_j \odot \boldsymbol{z}_k \right) \right\|^2 \tag{6}$$

$$L_{(C_i = C_j \sqcup C_k)} = \left\| \boldsymbol{y}_i - \max \left( \boldsymbol{y}_j, \boldsymbol{y}_k \right) \right\|^2 ; \ L_{(R_i = R_j \sqcup R_k)} = \left\| \boldsymbol{z}_i - \max \left( \boldsymbol{z}_j, \boldsymbol{z}_k \right) \right\|^2 \tag{7}$$

**Logical Negation** Necessary and sufficiency conditions are

$$L_{(C_i = \neg C_j)} = \left( \boldsymbol{y}_i^\top \boldsymbol{y}_j \right)^2 + \left( \overline{\boldsymbol{y}_i}^\top \overline{\boldsymbol{y}_j} \right)^2 ; \ L_{(R_i = \neg R_j)} = \left( \boldsymbol{z}_i^\top \boldsymbol{z}_j \right)^2 + \left( \overline{\boldsymbol{z}_i}^\top \overline{\boldsymbol{z}_j} \right)^2 \tag{8}$$

**Universal Type Restriction** For each given universal type restriction $\forall R_i \cdot C_j$, we need to ensure that for any $O_i, O_j \in \mathcal{N}_C$ having relation $R_i(O_i, O_j)$, we must have $O_j \in C_j$. A necessary and sufficient condition for this constraint would be as follows - whenever we have $O_j \in C_k$, where $C_k$ is a non-descendant of $C_j$ in the concept ontology, then we must not have $R_i(O_i, O_j)$. The loss term for this can be given as follows. For all $C_k \in \overline{\mathcal{D}}_{C_j}$, where $\overline{\mathcal{D}}_{C_j}$ denotes the set of concepts that are non-descendant of $C_j$, we can have the following loss term:

$$L_{(\forall R_i \cdot C_j)}(\boldsymbol{y}_k) = \left(\boldsymbol{y}_k^\top \left(\begin{smallmatrix} \mathbf{0}_d & \mathbf{I}_d \\ \mathbf{0}_d & \mathbf{0}_d \end{smallmatrix}\right) \boldsymbol{z}_i\right)^2 \tag{9}$$

The set $\overline{\mathcal{D}}_{C_j}$ can be shrunk considerably by considering only the following concepts in the concept hierarchy - (i) any child $C_k$ of the root concept $\top$ that is non-ancestor of $C_j$ (children of $C_k$ would automatically be taken care of due to quantum logical constraints related to inclusion), (ii) concept $C_k$ that is either a sibling of $C_j$ or child of an ancestor but itself is not an ancestor.

By putting all these losses together, we get the following unconstrained non-convex optimization problem as E2R learning problem and we call it as *E2R* model. Here, $L_{E2R}$ is the sum of all the loss term (each averaged over its corresponding logical assertions) defined from Equation (1) through (9).

$$\underset{\boldsymbol{x}_i, \boldsymbol{y}_i, \boldsymbol{z}_i}{\text{minimize}} \quad L_{E2R} \tag{10}$$

**Choosing the Embedding Dimension $d$** Under axis parallel assumption, we have a maximum of $2^{2d}$ different axis parallel subspaces when we embed into $\mathbb{C}^d$ (field being $\mathbb{R}$). Therefore, if we have $|\mathcal{N}_C|, |\mathcal{N}_R|$ many unique concepts and relations then we must have $d > \log(\sqrt{|\mathcal{N}_C| + |\mathcal{N}_R|})$. This is just a trivial lower bound, but in practice, we require much higher $d$ to accommodate all the constraints (especially regularity constraints given below). If we allow oblique subspace, we can embed in much smaller dimensional space. However, we still prefer axis-parallel embeddings because the distributive law, which in general doesn't hold true for quantum embeddings, surprisingly holds true under axis-parallel quantum embeddings (proof given in the supplementary material).

**The Problem of Subspace Collapse** The E2R formulation (10) is susceptible to one serious problem, namely *subspace collapse*. Imagine, there are *no logical negation and universal type restriction statements* in the input KB. In such a case, the loss terms (8), (9) would be missing and there would be a degenerate solution, namely *all the subspaces (i.e. $\boldsymbol{y}_i$ and $\boldsymbol{z}_i$) being the same*, because it would drive all the loss terms related to A-Box and T-Box assertions, namely (4) through (7), to zero. We can avoid this by including two kinds of regularity constraints - (i) For any two predicates (unary/binary) that are siblings of each other, we should encourage their subspaces to be orthogonal as much as possible, (ii) Any two predicates (unary/binary) that have (parent, child) relationship in the T-Box hierarchy, we should encourage certain minimum gap between the dimensions of their subspaces. Both these constraints can be achieved via the following losses.

$$\Omega_{\text{orth}} = \frac{1}{N_{\text{sib}}} \sum_{C_i \text{ sib } C_j} \left(\boldsymbol{y}_i^\top \boldsymbol{y}_j\right)^2 + \frac{1}{M_{\text{sib}}} \sum_{R_i \text{ sib } R_j} \left(\boldsymbol{z}_i^\top \boldsymbol{z}_j\right)^2 \tag{11}$$

$$\Omega_{\text{sep}} = \frac{1}{N_\sqsubseteq} \sum_{C_i \sqsubseteq C_j} \left(\sqrt{d} - \left(\boldsymbol{y}_j - \boldsymbol{y}_i\right)^\top \mathbf{1}\right)^2 + \frac{1}{M_\sqsubseteq} \sum_{R_i \sqsubseteq R_j} \left(\sqrt{d} - \left(\boldsymbol{z}_j - \boldsymbol{z}_i\right)^\top \mathbf{1}\right)^2 \tag{12}$$

The pairs $(C_i \text{ sib } C_j)$ and $(R_i \text{ sib } R_j)$ denote the pair of sibling concepts/relations. $N_{\text{sib}}$ and $M_{\text{sib}}$ denote the count of such sibling pairs. $N_\sqsubseteq$ and $M_\sqsubseteq$ denote the counts of respective logical assertions. The choice of $\sqrt{d}$ is flexible and can be replaced with an appropriate positive constant $\geq 1$. This choice allows us to stuff roughly $\sqrt{d}$ levels of the inclusion hierarchy which usually suffices.

Instead of using regularization losses (11) and (12), one can achieve a similar effect by having an alternative loss term which is simpler and effective in practice (we witnessed this in our experiments). This alternative loss term works as follows – Take each valid membership assertion in the given A-box. Replace one of its entity with an invalid (aka negatively sampled) entity. Take the negation of the corresponding membership loss of this newly constructed fake assertion (aka negative sample). In fact, this way of regularizing helps us in achieving good convergence during our model training. Note, we use a tiny fraction of invalid entities just to avoid degenerate solution. This should not be confused with closed-world assumption because there one uses a huge number of negative assertions.

**Reasoning Tasks**  While applicability of quantum embeddings is quite broad, in what follows, we highlight a few reasoning tasks that we think are important and apt for quantum embeddings. Each of these tasks could be deductive or predictive in nature and our approach need not even know this.

1. **Membership** [Find all the entities belonging the concept $C$ where $C$ could be a complex logical formula.] For this, we convert the formula $C$ into an appropriate subspace $S$, followed by finding all those entities whose length of orthogonal projection onto $S$ is more than some threshold. The projection length is used to assign the confidence score/probability of the answer being correct.

2. **Property Testing** [Does entity $O_i$ belongs to the concept $C$ where $C$ could be a complex logical formula?] The previous trick can be applied here as well and the answer would be probabilistic which can be made deterministic by setting an appropriate threshold.

3. **Property Listing** [Given entity $O_i$, list all the atomic concepts to which it belongs.] Find a set, say $B_i$, of all those standard basis vectors which are non-orthogonal to the embedding vector $x_i$ of $O_i$. Find all the atomic concepts $C_j$'s whose subspaces lie within the space of $B_i$. Each such $C_j$ would be an answer with confidence/probability proportional to the projection length.

## 5  Experiments

**Datasets and Setup**  We evaluated the performance of E2R on two different kinds of tasks – (i) *link prediction task*, and (ii) *reasoning task*. For link prediction, we chose FB15K and WN18 datasets because they are standard in the literature [5, 6, 24]. FB15K is a subset of the Freebase dataset, whereas WN18 is a subset of the WordNet dataset featuring lexical relations between words. The train and test sets of these datasets are respectively used for training and testing our proposed model.

To evaluate the reasoning capabilities, we chose LUBM (Lehigh University Benchmark) dataset (http://swat.cse.lehigh.edu/projects/lubm/), which consists of a university domain ontology, customizable and repeatable synthetic data. We generated one university (LUBM1U) data for the evaluation. We used all the 69628 triples produced by the LUBM generator along with ontology in triple form (unary and binary predicates) as our training set. Its important to mention that majority of the A-Box assertions in LUBM dataset are concerned about the bottom most predicates (concepts and relations) in the hierarchy. Starting with these assertions, we *custom designed* eight new *test queries* that comprise higher level predicates in the LUBM hierarchy so that answering such queries would explicitly require deductive reasoning. These test queries involved deducing the members of the following unary predicates – Professor, Faculty, Person, Student, Course, Organization, and the following binary predicates MemberOf, WorksFor.

We implemented E2R model using PyTorch. We used SGD (Stochastic Gradient Descent) with ADAM optimizer [25] to solve the proposed E2R formation (10) together with the regularization terms as described in Section 4. In all our experiments we used $d = 100$ for E2R model. E2R did not show significant sensitivity to the value of $d$, in our experimental range of 100 to 300. As specified in Section 4, the use of negative membership candidates during training helped in achieving good convergence. Using more than one negative candidate per positive membership candidate achieved better convergence. We used 3 different negative entities per positive entity in our experimental setup. For baseline approaches, we used optimal parameter setup obtained through grid search. For example, we used embedding dimension $d = 100$. Note, the datasets FB15k and WN18 do not contain T-Box. Therefore, we discarded the loss terms corresponding to T-Box assertions and retained the loss terms corresponding to A-Box assertions while training E2R on these datasets. Similarly, while training the baseline approaches on LUBM1U dataset, we simply added T-box assertions as additional triples because these approaches have no provision for handling T-box assertions in an explicit manner. Our experiments were performed on a Tesla K80 GPU machine.

**Baselines, Evaluation Metrics, and Results**  We used *TransE* [5] and *ComplEx* [24] as baselines to illustrate the effectiveness of our proposed model. TransE is a simple but effective model and ComplEx is one of the current state-of-the-art approach. We used OpenKE (https://github.com/thunlp/OpenKE) implementation of these approaches for our evaluation. We compared E2R with these approaches both for link prediction task (using FB15K and WN18 datasets) and reasoning tasks (using LUBM1U dataset). Tuning of the hyper-parameters for the baseline approaches was performed on the test set for FB15K and WN18 datasets but on the training set for LUBM1U. For E2R, the tuning was always done on the training set.

| Data | MEAN RANK | | | MRR | | | HITS@1 (%) | | | HITS@10 (%) | | |
|---|---|---|---|---|---|---|---|---|---|---|---|---|
| | $E2R$ | TE | CE | E2R | TE | CE | E2R | TE | CE | E2R | TE | CE |
| FB15K | 72.0 | 68.4 | 114.0 | **0.96** | 0.49 | 0.61 | **96.4** | 34.8 | 49.8 | **96.4** | 76.7 | 81.2 |
| WN18 | 5780.2 | 409.9 | 468.1 | 0.71 | 0.63 | 0.90 | 71.1 | 41.0 | 87.4 | 71.1 | 93.2 | 95.25 |
| LUBM1U | **220.1** | 1292.6 | 5742.9 | **0.46** | 0.26 | 0.12 | **45.4** | 18.97 | 12.5 | 45.4 | 49.1 | 12.59 |

**Table 1:** Datasets and Experimental Results. Acronyms used are TE := TransE, CE := ComplEx. LUBM-IU dataset has 69628 triples in A-Box, and 43 concepts and 25 relations in T-Box.

For all the evaluations, we used four standard metrics, namely MEAN RANK (lower the better), MRR (Mean Reciprocal Rank), HITS@*1*, and HITS@*10* [5]. These metrics capture how well an approach is able to extract correct entities for a given membership query. These metrics start improving as the correct answer entities start moving up in the ranked list of the predicted answer entities. Additionally, while computing these metrics, we considered only *filtered* case (as advocated in [5]) that avoid conflict among multiple correct answers in terms of occupying top rank. For reasoning task on LUBM1U dataset, where our goal is to evaluate the inferred membership of the concepts and relations, we computed all the metrics at the query level. The query level metric is an average over instance level metrics, where instances are the true members of the query. We further averaged these metrics over all the queries. Note, we did not use any symbolic reasoner as a baseline because our objective is not to compete with them but to demonstrate logical reasoning capability in vector spaces.

In E2R, the predicted candidate ranked list for any query is generated simply by projecting each entity onto the respective subspace and assigning the length of its residual component as its non-fitment score. For TransE/ ComplEx, a membership query rank orders the candidate triples based on the triple scores obtained through the corresponding model.

**Insights**    Table 1 shows experimental results, where we highlighted the numbers reflecting superior performance of E2R. It is evident from the results that E2R outperforms both TransE and ComplEx on metrics MRR and HITS@1 for all the tasks and datasets, except for WN18. Interestingly, E2R achieves an exceptional accuracy improvements over the baseline approaches for the link prediction task on the FB15k dataset as well as on the LUBM1U dataset for the reasoning task. This illustrates the effectiveness of E2R in learning and representing the logical structure of the input KB, and leveraging the same for the reasoning task. On the other hand, baseline approaches, which are fundamentally designed to address the link prediction task, find it hard to achieve the same. As far as WN18 dataset is concerned, its a collection of binary relation instances where most of these relations satisfy transitivity property (e.g. hypernym) and have inverse relations (e.g. hypernym/hyponym). Baseline methods, which are primarily distanced based, can probably capture transitivity/inversion properties bit better which E2R somewhat struggles because it is not distance based. Lastly, E2R consistently maintains a high score for HITS@1 metric and moreover, HITS@1 = HITS@10 for all the datasets. This implies that our algorithm often ranks a ground truth entity either at Rank-1 or at a quite low rank. We believe, this is an artifact of our approach trying to assign different sub-spaces for different predicates and pushing the non-members away from them. This phenomenon also results in a poor score for (average) MEAN RANK despite having high Hits@1 score. Such a behavior of E2R makes it practically interesting because Rank 1 candidate can be chosen with more confidence.

## 6   Related Work

- **[Quantum Logic]** Dominic [26, 21] advocated using QL framework to fix the problem of word meaning disambiguation in keyword based search engines. The conceptual framework suggested by Dominic and the follow-up work, however, do not prescribe any recipe to embed knowledge into vector space satisfying QL axioms. Our work aims to bridge this gap.

- **[Deep Nets for Word Embeddings]** In recent times, deep-net based techniques have emerged that can embed word-phrases [2, 27] and natural language sentences [28, 29] into a vector space. Such embeddings have offered remarkable performance boost for several downstream tasks including question answering [30, 31]. However, a key concern about these techniques is their black box nature which makes the underlying models unexplainable and non-interpretable.

- **[Statistical Relational Learning (SRL) and KG Embeddings]** For majority of the SRL techniques, the goal is to embed a *Knowledge Graph (KG)* into a vector space followed by discovery

of new triples or links [7, 32, 12]. The prediction of the links is done via a scoring function that builds upon exploiting the statistical measures for computing the similarity between joining entities. While effective for the KG completion tasks, this simple form of reasoning – based on the similarities among entities – face difficulties when handing complex reasoning queries [11]. One striking difference between E2R and KG embedding approaches is that later ones are not capable of handling the ontology of relations/concepts unlike E2R.

- **[Hierarchy Aware KB Embedding]** Recently, there have been attempts to embed ontological hierarchy [33, 34, 35, 36, 10] as well as inferring ontological hierarchy from embeddings [37, 38]. The work in [33] is less about embedding a KB and more about inducing an ontology. [34] embeds entities and predicates both in the forms of vectors which is very different from our philosophy. Unlike us, in [35], the training set upfront includes the assertions inferred on a select set of LUBM predicates (using symbolic reasoner) because their aim is to compete with symbolic reasoners (in speed and memory) during inference. [36] fall under the category of distance translation based embeddings (e.g. TransE) except that it uses 'type information' to improve quality of entities and relations embedding. Our Hits@10 score outperforms them on FB15K. In [10], they embed entities based on the type similarities as well as binary relations but they don't explicitly preserve ontology of binary relations themselves. In addition to the ontology embedding, there has also been some work on embedding the logic structure [39, 40, 41] of a KB into a vector space so as to enable reasoning. Unlike QL axioms in our approach, the underlying principles in all of these previous works do not offer any guarantee to preserve logical structure in the embedding space.

## 7   Conclusions

A novel approach, namely Embed2Reason, has been proposed which can embed a DL based symbolic KB into a vector space while preserving the structure of all logical propositions. The idea is inspired from Quantum Logic axioms. The key advantage is to be able to answer complex reasoning queries (both deductive and predictive) in accurate manner using distributional representation of the KB. The experimental results illustrate the effectiveness of E2R relative to standard baselines. The future directions include extension to more expressive logic as well as natural languages KBs.

**Acknowledgments**

We would like to thank Mukuntha N.S. for pointing out corrections at few places.

## Footnotes

*The first two authors contributed equally.

†This work was done when the author was with IBM Research, New Delhi, India.

[3]A similar motivation was advocated by [21] in the context of designing a document retrieval system.

[4]In general, QL allows the embedding space to be any finite or infinite dimensional Hilbert space.

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
