[Supplementary Material · Quantum_Embedding_Supplementary_Material_V1.pdf]

# Quantum Embedding of Knowledge for Reasoning (Supplementary Material)

**Dinesh Garg**[1*], **Shajith Ikbal**[1*], **Santosh K. Srivastava**[1], **Harit Vishwakarma**[2†],
**Hima Karanam**[1], **L Venkata Subramaniam**[1]

[1]IBM Research AI, India
[2]Dept. of Computer Sciences, University of Wisconsin-Madison, USA
garg.dinesh, shajmoha, sasriva5@in.ibm.com, hvishwakarma@cs.wisc.edu, hkaranam, lvsubram@in.ibm.com

## 1   Formal Concept Analysis (FCA) and Distributional Representation

FCA [1] is a branch of lattice theory [2] focusing on special type of lattices that are obtained from a set of *entities*, a set of *attributes*, and a binary relation called *context* that determines whether an entity has an attribute. The *context* relation is visualized by means of a table (called context-table) whose rows and columns correspond to entities and attributes, respectively, and entries are $1/0$ depending on whether an entity has an attribute or not (see Table 1 for an example). Although not directly

| ENTITY | $C_1$ | $C_2$ | $C_3$ |
|--------|-------|-------|-------|
| $O_1$  | 0     | 0     | 0     |
| $O_2$  | 0     | 0     | 1     |
| $O_3$  | 0     | 1     | 0     |
| $O_4$  | 0     | 1     | 1     |
| $O_5$  | 1     | 0     | 0     |
| $O_6$  | 1     | 0     | 1     |
| $O_7$  | 1     | 1     | 0     |
| $O_8$  | 1     | 1     | 1     |

**Table 1:** An illustrative FCA context-table

**Figure 1:** Embedding of Table 1 into vector space $F = \{0, 1\}^3$

related, FCA has lots of similarities with DL which can be exploited to get a naive and partial solution of the our central question in this paper, namely how to embed a DL based KB while preserving logical structure. This naive and partial solution also paves a way for us to think about quantum embeddings. To get such a naive and partial solution, given an A-Box, one needs to induce an FCA context-table as follows. Map all the entities and the leaf-level concepts (that is, bottom most concepts in the concept hierarchy) present in the A-Box to the entities and the attributes in FCA, respectively. If entity $O_i$ belongs to the leaf concept $C_j$ in the A-Box then mark 1 in the $(i, j)^{th}$-cell of the FCA context table, and 0 otherwise [3] Note, *any row of such induced FCA context-table can be treated as distributional representation or embedding* of the corresponding entity into a binary space $F = \{0, 1\}^d$, where $d$ is the number of leaf-level concepts in the A-Box.[4] Figure 1 depicts such an embedding for the entities present in the FCA context-table 1, where $d = 3$ and $m = 8$. A

vertex $v_i$ (red dot) in this embedding corresponds to a unique way of assigning membership of an entity across all the leaf-level concepts. Such an embedding preserves the logical structure of unary predicates as follows. Any atomic (or compound) concept of the given DL can be represented via a specific subset of these red vertices. For example, consider the compound concepts $C_4 = (C_2 \sqcap C_3)$ and $C_5 = (C_1) \sqcap (C_2 \sqcup C_3) \sqcap (\neg C_2 \sqcup \neg C_3)$. As shown in the Figure 1, these compound concepts can equivalently be represented via the vertex sets $\{v_4, v_8\}$ and $\{v_6, v_7\}$, respectively. In the same vein, one can convince that atomic concept $C_1$ can equivalently be represented via the vertex set $\{v_5, v_6, v_7, v_8\}$. For any concept $C \in \mathcal{C}$, we denote by $V(C)$ the set of vertices in $F$ which represents $C$ and by $Ext(C)$ its *extent - the set of all entities belonging of this concept* [1].

Given concepts $C_i$ and $C_j$, the logical inclusion $C_i \sqsubseteq C_j$ would be true iff concept $C_i$ is a specialization of concept $C_j$. This means any entity belonging to the concept $C_i$ would also belong to the concept $C_j$. Equivalently, $Ext(C_i) \subseteq Ext(C_j) \Leftrightarrow V(C_i) \subseteq V(C_j) \Leftrightarrow C_i \sqsubseteq C_j$. The logical inclusion $\sqsubseteq$ induces a partial order over the space of concepts $\mathcal{C}$. Such a partial order is also a lattice, where *unique meet (i.e. unique greatest lower bound)* and *unique join (i.e. unique least upper bound)* operations correspond to the logical union and intersection operations [1, 3]. That is, $C_i \vee C_j := C_i \sqcup C_j; C_i \wedge C_j := C_i \sqcap C_j$. In such a lattice, two concepts $C_i$ and $C_j$ are said to be negation of each other iff $C_i \vee C_j = \top$ and $C_i \wedge C_j = \bot$ [2]. If for every concept $C$, there is a unique complement, denoted by $\neg C$, then such a lattice is called as *orthocomplemented lattice* and serves as a device for *Boolean Logic* to process unary predicates based queries.

## 2 Distributive Law of Quantum Logic

The distributive law allows one to reformulate union and intersection of subspaces in logical proofs. Unlike the Boolean logic, the distributive law of subspaces does not hold true in general, and in particular to the Quantum logic. However, if the commutativity of the orthogonal projectors holds, then the distributive law of subspaces holds true. Here we prove that in axis-parallel embedding the commutativity of the orthogonal projectors hold and therefore the distributive law of subspaces follows. In order to prove this property, we need some facts about projection matrices which are encapsulated in following theorems.

**Theorem 1. Intersection of Subspaces [4]:** *Let $\boldsymbol{P}_1$ and $\boldsymbol{P}_2$ be the orthogonal projection matrices for the subspaces $S_1$ and $S_2$ respectively. In general, the subspaces $S_1$ and $S_2$ need not be disjoint. The necessary and sufficient condition for the matrix $\boldsymbol{P}_1\boldsymbol{P}_2$ to be an orthogonal projection matrix for the subspace $S_1 \sqcap S_2$ is*

$$\boldsymbol{P}_1\boldsymbol{P}_2 = \boldsymbol{P}_2\boldsymbol{P}_1. \tag{1}$$

**Theorem 2. Inclusion of Subspace:** *Let $\boldsymbol{P}_1$ and $\boldsymbol{P}_2$ be the orthogonal projection matrices for the subspaces $S_1$ and $S_2$, respectively. The necessary and sufficient condition for the us to have $S_1 \sqsubset S_2$ is*

$$\boldsymbol{P}_1\boldsymbol{P}_2 = \boldsymbol{P}_2\boldsymbol{P}_1 = \boldsymbol{P}_1. \tag{2}$$

**Proof of Theorem 2**
Since $S_1 \sqsubset S_2$, this implies $S_1 \sqcap S_2 = S_1$. The proof follows from Theorem 1.

**Corollary 3.** *Let $\boldsymbol{P}_1$ and $\boldsymbol{P}_2$ be the orthogonal projection matrices for the subspaces $S_1$ and $S_2$, respectively. If we have $S_1 \sqsubset S_2$, then for any $\boldsymbol{x} \in \mathbb{R}^d$, we must have*

$$\|\boldsymbol{P}_1\boldsymbol{x}\| \le \|\boldsymbol{P}_2\boldsymbol{x}\|. \tag{3}$$

**Corollary 4.** *Let $\boldsymbol{P}_1$ and $\boldsymbol{P}_2$ be the orthogonal projection matrices for the subspaces $S_1$ and $S_2$, respectively. The necessary and sufficient condition for $S_1$ and $S_2$ to be orthogonal subspaces is*

$$\boldsymbol{P}_1\boldsymbol{P}_2 = \boldsymbol{P}_2\boldsymbol{P}_1 = \boldsymbol{O}. \tag{4}$$

**Theorem 5. Union of Subspaces [4]:** *Let $\boldsymbol{P}_1$ and $\boldsymbol{P}_2$ be the orthogonal projection matrices for the subspaces $S_1$ and $S_2$, respectively, and let $\boldsymbol{P}_{1+2}$ denote the orthogonal projection matrix for the subspace $S_{1+2} = S_1 + S_2$. Then the following statements are equivalent:*

1. *$\boldsymbol{P}_1\boldsymbol{P}_2 = \boldsymbol{P}_2\boldsymbol{P}_1$.*

2. *$\boldsymbol{P}_{1+2} = \boldsymbol{P}_1 + \boldsymbol{P}_2 - \boldsymbol{P}_1\boldsymbol{P}_2$.*

**Corollary 6.** *Let $\boldsymbol{P}$ denote the orthogonal projection matrix for the subspace $S = S_1 + S_2$, and let $\boldsymbol{P}_1$, $\boldsymbol{P}_2$ be the orthogonal projection matrices for the subspaces $S_1$ and $S_2$ respectively. If $S_1$ and $S_2$ are orthogonal, then*

$$\boldsymbol{P} = \boldsymbol{P}_1 + \boldsymbol{P}_2.$$

**Lemma 7.** *Let $\boldsymbol{P}_1$, $\boldsymbol{P}_2$, and $\boldsymbol{P}_3$ be the orthogonal projection matrix onto the subspaces $S_1$, $S_2$, and $S_3$ respectively, and let $\boldsymbol{P}_{1+2}$ and $\boldsymbol{P}_{1+3}$ denote the orthogonal projection matrix onto the subspace $S_{1+2} = S_1 + S_2$ and $S_{1+3} = S_1 + S_3$. respectively. If $\boldsymbol{P}_1\boldsymbol{P}_2 = \boldsymbol{P}_2\boldsymbol{P}_2$, $\boldsymbol{P}_2\boldsymbol{P}_3 = \boldsymbol{P}_3\boldsymbol{P}_2$, and $\boldsymbol{P}_1\boldsymbol{P}_3 = \boldsymbol{P}_3\boldsymbol{P}_1$, then*

$$\boldsymbol{P}_{1+2}\boldsymbol{P}_{1+3} = \boldsymbol{P}_{1+3}\boldsymbol{P}_{1+2}. \tag{5}$$

**Proof of Lemma 7**

$$
\begin{aligned}
\boldsymbol{P}_{1+2}\boldsymbol{P}_{1+3} &= \left(\boldsymbol{P}_1 + \boldsymbol{P}_2 - \boldsymbol{P}_1\boldsymbol{P}_2\right)\left(\boldsymbol{P}_1 + \boldsymbol{P}_3 - \boldsymbol{P}_1\boldsymbol{P}_3\right) \\
&= \underbrace{\boldsymbol{P}_1^2 + \boldsymbol{P}_1\boldsymbol{P}_3 - \boldsymbol{P}_1^2\boldsymbol{P}_3}_{A} + \underbrace{\boldsymbol{P}_2\boldsymbol{P}_1 + \boldsymbol{P}_2\boldsymbol{P}_3 - \boldsymbol{P}_2\boldsymbol{P}_1\boldsymbol{P}_3}_{B} - \underbrace{\boldsymbol{P}_1\boldsymbol{P}_2\boldsymbol{P}_1 - \boldsymbol{P}_1\boldsymbol{P}_2\boldsymbol{P}_3 + \boldsymbol{P}_1\boldsymbol{P}_2\boldsymbol{P}_1\boldsymbol{P}_3}_{C}.
\end{aligned} \tag{6}
$$

The $A$ part of (6) simplifies to

$$
\begin{aligned}
A &= \boldsymbol{P}_1^2 + \boldsymbol{P}_1\boldsymbol{P}_3 - \boldsymbol{P}_1\boldsymbol{P}_1\boldsymbol{P}_3 \\
&\overset{(a)}{=} \boldsymbol{P}_1^2 + \boldsymbol{P}_3\boldsymbol{P}_1 - \boldsymbol{P}_1\boldsymbol{P}_3\boldsymbol{P}_1 \\
&= \left(\boldsymbol{P}_1 + \boldsymbol{P}_3 - \boldsymbol{P}_1\boldsymbol{P}_3\right)\boldsymbol{P}_1,
\end{aligned} \tag{7}
$$

where in $(a)$ we used $\boldsymbol{P}_1\boldsymbol{P}_3 = \boldsymbol{P}_3\boldsymbol{P}_1$ in the $2^{\text{th}}$ and $3^{\text{rd}}$ terms. The $B$ part of (6) simplifies to

$$
\begin{aligned}
B &= \boldsymbol{P}_2\boldsymbol{P}_1 + \boldsymbol{P}_2\boldsymbol{P}_3 - \boldsymbol{P}_2\boldsymbol{P}_1\boldsymbol{P}_3 \\
&\overset{(b)}{=} \boldsymbol{P}_1\boldsymbol{P}_2 + \boldsymbol{P}_2\boldsymbol{P}_3 - \boldsymbol{P}_1\boldsymbol{P}_2\boldsymbol{P}_3 \\
&\overset{(c)}{=} \boldsymbol{P}_1\boldsymbol{P}_2 + \boldsymbol{P}_3\boldsymbol{P}_2 - \boldsymbol{P}_1\boldsymbol{P}_3\boldsymbol{P}_2 \\
&= \left(\boldsymbol{P}_1 + \boldsymbol{P}_3 - \boldsymbol{P}_1\boldsymbol{P}_3\right)\boldsymbol{P}_2,
\end{aligned} \tag{8}
$$

where in $(b)$ we used $\boldsymbol{P}_2\boldsymbol{P}_1 = \boldsymbol{P}_1\boldsymbol{P}_2$ in the $1^{\text{st}}$ and the last term, while in $(c)$ we used $\boldsymbol{P}_2\boldsymbol{P}_3 = \boldsymbol{P}_3\boldsymbol{P}_2$ in the $2^{\text{nd}}$ and the last term. Finally, the $C$ part of (6) simplifies to

$$
\begin{aligned}
C &= -\boldsymbol{P}_1\boldsymbol{P}_2\boldsymbol{P}_1 - \boldsymbol{P}_1\boldsymbol{P}_2\boldsymbol{P}_3 + \boldsymbol{P}_1\boldsymbol{P}_2\boldsymbol{P}_1\boldsymbol{P}_3 \\
&\overset{(d)}{=} -\boldsymbol{P}_1\boldsymbol{P}_1\boldsymbol{P}_2 - \boldsymbol{P}_1\boldsymbol{P}_2\boldsymbol{P}_3 + \boldsymbol{P}_1\boldsymbol{P}_1\boldsymbol{P}_2\boldsymbol{P}_3 \\
&\overset{(e)}{=} -\boldsymbol{P}_1\boldsymbol{P}_1\boldsymbol{P}_2 - \boldsymbol{P}_1\boldsymbol{P}_3\boldsymbol{P}_2 + \boldsymbol{P}_1\boldsymbol{P}_1\boldsymbol{P}_3\boldsymbol{P}_2 \\
&\overset{(f)}{=} -\boldsymbol{P}_1\boldsymbol{P}_1\boldsymbol{P}_2 - \boldsymbol{P}_3\boldsymbol{P}_1\boldsymbol{P}_2 + \boldsymbol{P}_1\boldsymbol{P}_3\boldsymbol{P}_1\boldsymbol{P}_2 \\
&= -\left(\boldsymbol{P}_1 + \boldsymbol{P}_3 - \boldsymbol{P}_1\boldsymbol{P}_3\right)\boldsymbol{P}_1\boldsymbol{P}_2,
\end{aligned} \tag{9}
$$

where in $(d)$ we used $\boldsymbol{P}_2\boldsymbol{P}_1 = \boldsymbol{P}_1\boldsymbol{P}_2$ to the $1^{\text{st}}$ and the last term, in $(e)$ we used $\boldsymbol{P}_2\boldsymbol{P}_3 = \boldsymbol{P}_3\boldsymbol{P}_2$ to the $2^{\text{nd}}$ and the last term, and in $(f)$ we used $\boldsymbol{P}_1\boldsymbol{P}_3 = \boldsymbol{P}_3\boldsymbol{P}_1$ to the $2^{\text{nd}}$ and the last term. Using (7), (8), and (9) in (6), we have

$$
\begin{aligned}
\boldsymbol{P}_{1+2}\boldsymbol{P}_{1+3} &= \left(\boldsymbol{P}_1 + \boldsymbol{P}_3 - \boldsymbol{P}_1\boldsymbol{P}_3\right)\boldsymbol{P}_1 + \left(\boldsymbol{P}_1 + \boldsymbol{P}_3 - \boldsymbol{P}_1\boldsymbol{P}_3\right)\boldsymbol{P}_2 - \left(\boldsymbol{P}_1 + \boldsymbol{P}_3 - \boldsymbol{P}_1\boldsymbol{P}_3\right)\boldsymbol{P}_1\boldsymbol{P}_2 \\
&= \left(\boldsymbol{P}_1 + \boldsymbol{P}_3 - \boldsymbol{P}_1\boldsymbol{P}_3\right)\left(\boldsymbol{P}_1 + \boldsymbol{P}_2 - \boldsymbol{P}_1\boldsymbol{P}_2\right) \\
&= \boldsymbol{P}_{1+3}\boldsymbol{P}_{1+2}.
\end{aligned} \tag{10}
$$

**Theorem 8. Distributive Law of Subspaces:** *Let $\boldsymbol{P}_1, \boldsymbol{P}_2, \boldsymbol{P}_3$ denote the orthogonal projection matrices for the subspaces $S_1, S_2, S_3$, respectively. If $\boldsymbol{P}_1\boldsymbol{P}_2 = \boldsymbol{P}_2\boldsymbol{P}_1$, $\boldsymbol{P}_2\boldsymbol{P}_3 = \boldsymbol{P}_3\boldsymbol{P}_2$, and $\boldsymbol{P}_1\boldsymbol{P}_3 = \boldsymbol{P}_3\boldsymbol{P}_1$, then the following relations of distributive law of subspaces hold:*

$$
\begin{aligned}
S_1 + (S_2 \sqcap S_3) &= (S_1 + S_2) \sqcap (S_1 + S_3), & (11) \\
S_2 + (S_1 \sqcap S_3) &= (S_1 + S_2) \sqcap (S_2 + S_3), & (12) \\
S_3 + (S_1 \sqcap S_2) &= (S_1 + S_3) \sqcap (S_2 + S_3). & (13)
\end{aligned}
$$

**Proof of Theorem 8**

We will prove (11) and the proof of (12) and (13) are similar. Since there is a one-to-one correspondence between projectors and and subspaces, we show that the projection on the LHS subspace of (11) is equal to the projection on the RHS subspace. The orthogonal projector $\mathbf{P}_{2\sqcap 3}$ on the subspace $S_2 \sqcap S_3$ is given by $\mathbf{P}_{2\sqcap 3} = \mathbf{P}_2\mathbf{P}_3 = \mathbf{P}_3\mathbf{P}_2$.

Let $\mathbf{P}_{1+(2\sqcap 3)}$ denotes the orthogonal projection matrix for the subspace $S_1 + (S_2 \sqcap S_3)$. Then using Theorem 1 and 2, $\mathbf{P}_{1+(2\sqcap 3)}$ can be written as

$$
\begin{aligned}
\mathbf{P}_{1+(2\sqcap 3)} &= \mathbf{P}_1 + \mathbf{P}_{2\sqcap 3} - \mathbf{P}_1\mathbf{P}_{2\sqcap 3}, \\
&= \mathbf{P}_1 + \mathbf{P}_2\mathbf{P}_3 - \mathbf{P}_1\mathbf{P}_2\mathbf{P}_3.
\end{aligned}
\tag{14}
$$

The orthogonal projection matrices $\mathbf{P}_{1+2}$ and $\mathbf{P}_{1+3}$ for the subspaces $S_1+S_2$ and $S_1+S_3$, respectively are given by

$$
\begin{aligned}
\mathbf{P}_{1+2} &= \mathbf{P}_1 + \mathbf{P}_2 - \mathbf{P}_1\mathbf{P}_2, \\
\mathbf{P}_{1+3} &= \mathbf{P}_1 + \mathbf{P}_3 - \mathbf{P}_1\mathbf{P}_3.
\end{aligned}
$$

Using Lemma 7, it is easy to see that $\mathbf{P}_{1+2}\mathbf{P}_{1+3} = \mathbf{P}_{1+3}\mathbf{P}_{1+2}$ holds. Hence, the orthogonal projector onto the subspace $(S_1 + S_2) \sqcap (S_1 + S_3)$ is

$$
\begin{aligned}
\mathbf{P}_{1+2}\mathbf{P}_{1+3} &= (\mathbf{P}_1 + \mathbf{P}_2 - \mathbf{P}_1\mathbf{P}_2)(\mathbf{P}_1 + \mathbf{P}_3 - \mathbf{P}_1\mathbf{P}_3) \\
&\stackrel{(a)}{=} \mathbf{P}_1 + \mathbf{P}_2\mathbf{P}_1 + \mathbf{P}_2\mathbf{P}_3 - \mathbf{P}_2\mathbf{P}_1\mathbf{P}_3 - \mathbf{P}_1\mathbf{P}_2\mathbf{P}_1 - \mathbf{P}_1\mathbf{P}_2\mathbf{P}_3 + \mathbf{P}_1\mathbf{P}_2\mathbf{P}_1\mathbf{P}_3 \\
&\stackrel{(b)}{=} \mathbf{P}_1 + \mathbf{P}_1\mathbf{P}_2 + \mathbf{P}_2\mathbf{P}_3 - 2\mathbf{P}_1\mathbf{P}_2\mathbf{P}_3 - \mathbf{P}_1^2\mathbf{P}_2 + \mathbf{P}_1^2\mathbf{P}_2\mathbf{P}_3 \\
&= \mathbf{P}_1 + \mathbf{P}_2\mathbf{P}_3 - \mathbf{P}_1\mathbf{P}_2\mathbf{P}_3 \\
&\stackrel{(c)}{=} \mathbf{P}_{1+(2\sqcap 3)}.
\end{aligned}
\tag{15}
$$

where, in $(a)$ we used the projection property $\mathbf{P}_1^2 = \mathbf{P}_1$, $(b)$ follows by replacing $\mathbf{P}_2\mathbf{P}_1$ with $\mathbf{P}_1\mathbf{P}_2$ in the $2^{nd}$, $4^{th}$, $5^{th}$ and the last terms of $(a)$, and finally, in $(c)$ we used (14).

**Theorem 9.** *Let $A = [e_1 | e_2 | \dots | e_m]$, where $e_1, e_2, \dots, e_m$ is subset of a standard basis vectors of $\mathbb{R}^d$. Then the orthonormal projector $P$ onto the subspace $S = \mathsf{span}(A)$ spanned by the basis vectors $e_1, e_2, \dots, e_m$ is given by*

$$
P = AA^T.
\tag{16}
$$

**Proof of Theorem 9**

The projection matrix onto the subspace $S = \mathsf{span}(A)$ is

$$
\begin{aligned}
\mathbf{P} &= \mathbf{A}\left(\mathbf{A}^T\mathbf{A}\right)^{-1}\mathbf{A}^T \\
&\stackrel{(d)}{=} \mathbf{A}\mathbf{I}_m\mathbf{A}^T = \mathbf{A}\mathbf{A}^T,
\end{aligned}
\tag{17}
$$

where $(d)$ follows, since columns of $\mathbf{A}$ are orthonormal, $\mathbf{A}^T\mathbf{A}$ is equal to m-by-m identity matrix $\mathbf{I}_m$.

**Theorem 10. Commutativity of Projections in Axis-Parallel Embedding:** *Let $A_1 = [e_{i_1} | e_{i_2} | \dots | e_{i_m}]$, and $A_2 = [e_{j_1} | e_{j_2} | \dots | e_{j_n}]$, where $e_{i_1}, e_{i_2}, \dots, e_{i_m}$ and $e_{j_1}, e_{j_2}, \dots, e_{j_n}$ are two subsets of a standard basis vectors in $\mathbb{R}^d$. If $P_1$ and $P_2$ be the orthogonal projector onto the subspace $S_1 = \mathsf{span}(A_1)$ and $S_2 = \mathsf{span}(A_2)$ respectively, then*

$$
P_1P_2 = P_2P_1 = \sum_{k=1}^{m}\sum_{l=1}^{n} \delta_{i_k, j_l}\left(e_{i_k}e_{j_l}^T\right),
\tag{18}
$$

*where*

$$
\delta_{i_k, j_l} = \begin{cases} 1 & \text{if } i_k = j_l \\ 0 & \text{otherwise} \end{cases}
\tag{19}
$$

**Proof of Theorem 10**

We prove (18) using matrix computation. Using Theorem (8)

$$
\begin{aligned}
\mathbf{P}_1\mathbf{P}_2 &= \mathbf{A}_1\left(\mathbf{A}_1^T\mathbf{A}_2\right)\mathbf{A}_2^T \\
&= \mathbf{A}_1\boldsymbol{\Delta}\mathbf{A}_2^T,
\end{aligned}
$$

where $\mathbf{\Delta}$ is m-by-n matrix

$$
\begin{aligned}
\mathbf{\Delta} &= \begin{bmatrix} \mathbf{e}_{i_1}^T \\ \mathbf{e}_{i_2}^T \\ \vdots \\ \mathbf{e}_{i_m}^T \end{bmatrix} \begin{bmatrix} \mathbf{e}_{j_1} | \mathbf{e}_{j_2} | & \cdots & | \mathbf{e}_{j_n} \end{bmatrix} \\
&= \begin{bmatrix} \delta_{i_1,j_1} & \delta_{i_1,j_2} & \cdots & \delta_{i_1,j_n} \\ \delta_{i_2,j_1} & \delta_{i_2,j_2} & \cdots & \delta_{i_2,j_n} \\ \vdots & \vdots & \vdots & \vdots \\ \delta_{i_m,j_1} & \delta_{i_m,j_2} & \cdots & \delta_{i_m,j_n} \end{bmatrix},
\end{aligned} \tag{20}
$$

where $ij^{\text{th}}$ entry of $\mathbf{\Delta}$ is given by (19). Thus the projection matrix $\mathbf{P}_1 \mathbf{P}_2$ becomes

$$
\begin{aligned}
\mathbf{P}_1 \mathbf{P}_2 &= \begin{bmatrix} \mathbf{e}_{i_1} | \mathbf{e}_{i_2} | & \cdots & | \mathbf{e}_{i_m} \end{bmatrix} \begin{bmatrix} \delta_{i_1,j_1} & \delta_{i_1,j_2} & \cdots & \delta_{i_1,j_n} \\ \delta_{i_2,j_1} & \delta_{i_2,j_2} & \cdots & \delta_{i_2,j_n} \\ \vdots & \vdots & \vdots & \vdots \\ \delta_{i_m,j_1} & \delta_{i_m,j_2} & \cdots & \delta_{i_m,j_n} \end{bmatrix} \begin{bmatrix} \mathbf{e}_{j_1}^T \\ \mathbf{e}_{j_2}^T \\ \vdots \\ \mathbf{e}_{j_n}^T \end{bmatrix} \\
&= \sum_{k=1}^m \sum_{l=1}^n \delta_{i_k,j_l} \mathbf{e}_{i_k} \mathbf{e}_{j_l}^T \\
&\overset{(e)}{=} \sum_{l=1}^n \sum_{k=1}^m \delta_{j_l,i_k} \mathbf{e}_{j_l} \mathbf{e}_{i_k}^T \\
&= \begin{bmatrix} \mathbf{e}_{j_1} | \mathbf{e}_{j_2} | & \cdots & | \mathbf{e}_{j_n} \end{bmatrix} \begin{bmatrix} \delta_{j_1,i_1} & \delta_{j_1,i_2} & \cdots & \delta_{j_1,i_m} \\ \delta_{j_2,i_1} & \delta_{j_2,i_2} & \cdots & \delta_{j_2,i_m} \\ \vdots & \vdots & \vdots & \vdots \\ \delta_{j_n,i_1} & \delta_{j_n,i_2} & \cdots & \delta_{j_n,i_m} \end{bmatrix} \begin{bmatrix} \mathbf{e}_{i_1}^T \\ \mathbf{e}_{i_2}^T \\ \vdots \\ \mathbf{e}_{i_m}^T \end{bmatrix} \\
&= \mathbf{A}_2 \mathbf{\Delta}^T \mathbf{A}_1^T, \\
&= \mathbf{P}_2 \mathbf{P}_1,
\end{aligned} \tag{21}
$$

where in $(e)$ taking a transpose holds because when $i_k \neq j_l$, $\mathbf{e}_{i_k} \mathbf{e}_{j_l}^T$ becomes a d-by-d zero matrix, when $i_k = j_l$, the outer product $\mathbf{e}_{i_k} \mathbf{e}_{j_l}^T = \mathbf{e}_{i_k} \mathbf{e}_{i_k}^T$ which becomes a d-by-d diagonal matrix.

**Theorem 11. Distributive Law of Subspace in Axis-Parallel Embedding:** *Let $A_1 = [\boldsymbol{e}_{i_1} | \boldsymbol{e}_{i_2} | \ldots | \boldsymbol{e}_{i_m}]$, $A_2 = [\boldsymbol{e}_{j_1} | \boldsymbol{e}_{j_2} | \ldots | \boldsymbol{e}_{j_n}]$, and $A_3 = [\boldsymbol{e}_{k_1} | \boldsymbol{e}_{k_2} | \ldots | \boldsymbol{e}_{k_r}]$, where $\boldsymbol{e}_{i_1}, \boldsymbol{e}_{i_2}, \ldots, \boldsymbol{e}_{i_m}$, $\boldsymbol{e}_{j_1}, \boldsymbol{e}_{j_2}, \ldots, \boldsymbol{e}_{j_n}$, and $\boldsymbol{e}_{k_1}, \boldsymbol{e}_{k_2}, \ldots, \boldsymbol{e}_{k_r}$ are three subsets of a standard basis vectors in $\mathbb{R}^d$. If $P_1$, $P_2$, and $P_3$ are the orthogonal projection matrices for the subspaces $S_1 = \mathsf{span}(A_1)$, $S_2 = \mathsf{span}(A_2)$, and $S_3 = \mathsf{span}(A_3)$, respectively, then*

$$
S_1 + (S_2 \sqcap S_3) = (S_1 + S_2) \sqcap (S_1 + S_3). \tag{22}
$$

**Proof of Theorem 11**
Using Theorem (9) we were able to show that $\mathbf{P}_1 \mathbf{P}_2 = \mathbf{P}_2 \mathbf{P}_1$. In a similar way, we can prove that $\mathbf{P}_2 \mathbf{P}_3 = \mathbf{P}_3 \mathbf{P}_2$, and $\mathbf{P}_1 \mathbf{P}_3 = \mathbf{P}_3 \mathbf{P}_1$. Because all the conditions of Theorem (7) hold, therefore the distributive law of subspaces (22) holds.

## Footnotes

*The first two authors contributed equally.

†This work was done when the author was with IBM Research, New Delhi, India.

[3]This is what called as *closed world assumption* and FCA works on this assumption.

[4]Its not obvious how to embed relations and entity pairs in the same space and hence it is a partial solution.