[Reviews · NeurIPS 2019]

Reviewer 1



The work introduces a beautiful formalism and a host of reasoning-related operations. For instance, they introduce the equivalent of basic logical operations (AND, OR, negation) in their formalism. Additionally, they formalize higher-level operations based on these atomic relations. For instance, enforcement of type-hierarchy in the representation or membership queries. My biggest concern is the experimental evaluations. The first experiment (link prediction) is a must here, but certainly not enough. The other experiments on the LUBM dataset, which is claimed to evaluate “reasoning capabilities” is a bit vague. In particular, when you say “we custom designed eight new queries that comprise higher level predicates,” I would like to understand it better how you did this. Interpreting the results in Table 1 is a bit hard: - When looking at HITS@1, E2R is way better than others in FB15K, but not much difference between E2R and CE in WN18 (in fact, CE is better); why? - Why in WN18, the mean rank of E2R is vastly bigger than other systems? Overall, it’s a nicely written work and I enjoyed reading work. I would have expected more careful experimentation on the claim, especially in more natural settings. Writing: - “aims to bridges”: to bridge - Figure 1, the label of the right figure should be “T-Box”? - In Table 1, the middle row you never boldface any of the systems; not sure if it was intentional (also the top-left row.) - “embed a DL based symbolic KB into Hilbert space”: we’re really mapping KB to finite vectors in R^d (which is a subset of Holbert space, but it does not cover the whole Hilbert space. Might be helpful to be clear about this, so that it doesn’t come off as an over-claim. - There are some terms borrowed from abstract algebra literature which could be clarified with an extra bit of explanation/pointers: “field”, C (space of complex numbers?), “isomorphic”. - There are other works that try to induce logical relations between the concepts/relations when embedding; for instance: * Low-Dimensional Embeddings of Logic, 2014 * Injecting Logical Background Knowledge into Embeddings for Relation Extraction, 2015 * Lifted Rule Injection for Relation Embeddings, 2016

Reviewer 2



The authors introduce a new way of embedding entities and relations into a R^n vector space via quantum logic. Quantum Logic provides the main framework by which they can show that, given an embedding that fits all the requirements, queries become a matter of taking the distance to a plane. Queries here are in the sense of description logic: set membership, containment, etc. The embedding is found through SGD with a loss function that represents all of the constraints the embedding needs to satisfy. The authors did an excellent job of providing the necessary background and walking through the construction and to the experiments. There were a few times I jotted down a question to ask, and then found it answered in the next paragraph. The writing is clear, though if there was space it would be nice to expand a bit to help the reader follow the equations 4--9. One question I didn't follow is how simple the operations in section 3 are. Is it trivial to intersect two subspaces, add two subspaces, and find the orthogonal complement of a subspace? Are the subspaces each represented with just a single 2d-dimensional vector, or is it more than that? Is it a span of a set of vectors, and these operations are trivial operations on those sets? Just looking for more understanding of the cost of these operations and the cost of representation in the space. One question I had is how deep of an ontology of relations can be represented with n-dimensional space. Is it n, or something larger? Can trinary or higher-order predicates be represented trivially by extending the space to R^3n for example, or does the entire technique support only unary and binary predicates? The description of the experiments makes it sound like the baseline system hyperparameters were tuned on the test set. Can you verify if that is true or if it was done more carefully with something like cross-validation or a separate development set? There seems to be a typo in Table 1 where the HITS@1 and HITS@10 results are the same for E2R. I'm hoping they are the results for HITS@1, but more likely are the actual results for HITS@10. Line 289-290 made me remember that there are other reasons for doing an embedding of the KB other than as an approximation to doing logic reasoning. For logic reasoning, using a symbolic reasoner will perform better (as stated). However, there are other benefits of an embedding. For instance, it may be useful for prediction (as you show with the link prediction task?). Just to confirm: A symbolic logic reasoner would perform poorly on the two link prediction tasks, right? Because the link prediction is based on some unknown fuzzy properties of the entities and relations they belong to, not simply a reasoning task, is that correct? Another benefit, (not evaluated here), is that usually one can say something about relation and entity similarity based on closeness of their vectors. A qualitative evaluation of this property may also lend value to the paper. I think a brief discussion of the pros and cons of symbolic reasoning vs. embedded would be interesting, informative, and also strengthen the paper to point out the benefits of what you've done. small typo: line 277 "predication" should be "prediction". ** Update after author response: Thank you for answering my questions in your author response.

Reviewer 3



The paper presents an interesting idea to learn embeddings of hierarchies and ontologies based on quantum logic. I find the proposed formulation to be novel and a positive contribution. However, I do not think that the experiments fully bear out the utility of the proposed approach. The loss function (10), if solved exactly, would capture the logical structure of hierarchies. However, because it is a non-convex problem, the paper solves it via stochastic gradient descent. Hence, there is no assurance that an optimal solution is always found, and thus the proposed approach suffers from the same problem (perhaps to a lesser degree) as other embedding approaches, i.e., "no guarantee that embeddings maintain the sanctity of the logical structure" (line 35-36). Because of this, I think it behooves the paper to compare itself against other embedding approaches that attempt to incorporate hierarchical structure (in addition to non-hierarchical approaches like TransE and ComplEx that were used as baselines in the paper). The citations on line 326 (i.e., [32, 33, 34, 35, 10]) provide a list of possible hierarchical embeddings approaches that could be compared against. Two additional ones are: (a) Poincaré Embeddings for Learning Hierarchical Representations. Maximilian Nickel, Douwe Kiela. NIPS, 2017. (b) Learning Continuous Hierarchies in the Lorentz Model of Hyperbolic Geometry. Maximilian Nickel, Douwe Kiela. ICML, 2018. I think it would have been instructive if the paper has compared against these "hierarchy-aware" approaches, because the comparisons would show how much better the paper's *approximate* quantum-logical constraints are versus other ways of embedding hierarchies. I think a running illustrative example of the components of equation 10 on page 4 and 5 would greatly improve the comprehensibility of the paper. Questions: 1. How does the paper's approach compare against (a) and (b) above? 2. Why did the paper not compare against the "hierarchy-aware" approaches mentioned above ([32, 33, 34, 35, 10])? 3. Page 6, line 222-227: By sampling invalid entities, isn't the paper implicitly also making the closed-world assumption, a shortcoming that the paper critiques on line 86-87? 4. Page 6, line 233: How does the paper "[find] all those entities"? By enumeration? 5. Page 6, line 242: How does confidence/probability correspond to projection length? There is no information about this in the paper or the supplementary material. 6. Page 7, Table 1: What exactly constitutes the "reasoning task" on LUBM? What must be inferred? How many steps of reasoning are required? How big is the training and test data? As it stands, this experiment is hard to replicate. 7. Page 7, Table 1: Why is the paper's approach not performing well on WN18? 8. Page 7, line 266, "better convergence": Why would you get better convergence with 3 negative entities per positive entity? What's a negative entity? 9. Page 8, line 291-293, "projecting each entity ... non-fitment score": Could the author elaborate on what this sentence means? 10. Page 9, line 315, "their black box nature": The paper's embedding is also a black-box, no? What interpretability does its embedding offer? Nits: * Page 4 line 134: implies following -> implies the following (same mistake present elsewhere in paper) * Page 5 line 189: sufficiency -> sufficient * Page 8 line 310: bridges -> bridge UPDATE: The authors' feedback addressed my concerns to a large extent (though I still feel more details won't hurt the reproducibility of the LUBM experiments). Hence, I've upgraded my overall score to "6: Marginally above the acceptance threshold".

[Author Response · NeurIPS 2019]

**All Reviewers:** Thanks for your insightful reviews. We have tried our best to clarify them below (common responses
followed by individual responses). [A cryptic form of reviewer question/comment precedes our response.]
[Experimental Results Interpretation] We confirm that Hits@1 and Hits@10 scores in Table 1 are correct. Note, it
implies that our algorithm often ranks a ground truth entity either at Rank 1 or at a quite low rank. We believe, this is an
artifact of our approach trying to assign different sub-spaces for different predicates and pushing the non-members away
from them. This phenomenon also results in a poor score for (average) mean rank despite having high Hits@1 score.
Such a behavior of E2R, as stated in paper, makes it practically interesting because Rank 1 candidate can be chosen
with more confidence. [WN18 Dataset] Its a collection of binary relation instances where most of these relations satisfy
transitivity property (e.g. hypernym) and have inverse relations (e.g. hypernym/hyponym). This dataset comes in the
form of A-box and the underlying T-box capturing transitivity/inversion information is not supplied. Baseline methods,
which are primarily distanced based, can probably capture transitivity/inversion properties from the A-box a bit better.
[Deduction on LUBM] Deduction task is to deduce (pair of) entities that belong to a non-leaf (relation) concept in
LUBM ontology. Each of 8 custom query is one such non-leaf (relation) concept – details at line no. 257-259 in paper.
Training set include 69628 triples. Number of reasoning steps required can be as many as tree depth which is 4 here.
**Reviewer #1** [Custom queries for LUBM] + [Results Interpretation] Please refer to line no. 3-14 above. ["T-Box" label
in Figure 1] The label of the right figure in Figure 1 is actually correct. Both left and right figures contain "A-box" (oval
nodes) as well as "T-box" (rectangle nodes) but the difference is that they depict unary and binary predicate hierarchy,
respectively. We will clarify it. [Boldfacing Middle row in Table 1] As mentioned at line no 295–296 in paper, we
chose to boldface only those entries which depict superior performance of E2R. We felt this may ease the comparison.
[Abstract algebra terms] Yes, the field $\mathbb{C}$ is space of complex numbers. We will clarify this as well as isomorphism.
[Hilbert Space vs. $R^d$]+ [Other work on inducing logical relation] Points well taken. We will address them.
**Reviewer #2** [Expanding equations 4–9] We will include an explanation. [Operations in Section 3] These operations
are trivial because we denote subspaces by indicator vectors and intersection, union, complement can simply be
implemented via bit-wise AND, OR, FLIP operations. [Ontology Depth] For $n$-dimensional space, ideally, the ontology
depth should be no more than $n$. [Higher-order Predicate] We also have a strong conviction that the trinary predicates
can be represented by $R^{3n}$. However, other details (e.g. inner product definition) need to be worked out for this setup
and that is our future direction. [Tuning of Baseline Parameters] Tuning of the parameters for the baseline approaches
was performed on the dev set for FB15K and WN18 but on the training set for LUBM. For E2R, the tuning was always
done on the training set. [HITS@1 and HITS@10 being same for E2R] Please refer to line no. 3–7 above. [Symbolic
logic for link prediction] Yes, symbolic reasoners are typically not designed for link prediction tasks. [Symbolic
reasoning vs. embedding] You are right in your assessment. We will surely include a discussion.
**Reviewer #3** [Optimality of loss funct. (10)] We agree that locally optimal embedding given by SGD may not fully
maintain the sanctity of the logical structure. However, because our formulation explicitly models sanctity, such a
local-optima would still respect the sanctity to a greater degree (if not 100%) and hence would likely be better than
that of the baseline approaches (as evident from LUBM results). Designing improved optimization technique is an
interesting future direction. [Illustrative Example] We will try running through the example of Figure 2 for better
explanation of eqn. (10). [Poincaré Embeddings + Lorentz Model] Thanks for pointing. By reading, we felt these
methods do not make use of hierarchy information even if it is explicitly supplied in the form of T-box. Instead, they
infer such hierarchical structure in an unsupervised manner. Therefore, these methods, by design, are not geared to
tackle deductive reasoning problems where T-box plays a major role. We will include a discussion. [Comparison with
32,33,34,35,10] [32] is less about embedding a KB and more about inducing an ontology. [33] embeds entities and
predicates both in the forms of vectors which is very different from our philosophy. Unlike us, in [34], the training set
upfront includes the assertions inferred on a select set of LUBM predicates (using symbolic reasoner) because their aim
is to compete with symbolic reasoners (in speed and memory) during inference. [35] fall under the category of distance
translation based embeddings (e.g. TransE) except that it uses 'type information' to improve quality of entities and
relations embedding. Our Hits@10 score outperforms them on FB15K. In [10], they embed entities based on the type
similarities as well as binary relations but they don't explicitly preserve ontology of binary relations themselves. We will
include these comparisons. [Sampling invalid entities vs. closed-world assumption] Like other approaches, we use a
tiny fraction of invalid entities just to avoid degenerate solution, where all predicates (entities) collapse to a single space
(vector). A full blown closed-world assumption involves many more negative assertions (e.g. missing edges among
predicates) which we never touched. ["reasoning task" on LUBM?] + [Performance on WN18] Please see line no. 3–14
above. [Finding all the entities] Yes, we enumerate all the entities. [Confidence/prob. corresponding to proj. length]+
["projecting each entity...non-fitment score"] Because E2R aspires to embed an entity within a concept subspace to
which it belongs, we felt a natural loss metric would be residual length of its projection. The same idea is also endorsed
by the Quantum Logic where projection length is indeed used as a probability of such membership assertion. We will
clarify it. [better convergence with 3 -Ve entities] -Ve entities are defined at line no. 222-227 in paper and are used to
improve convergence. Number of -Ve entities per +Ve entity is a hyper-parameter and value of 3 gave the best training
accuracy. [Interpretability] Our formulation has separate loss for each aspect of preserving logical sanctity ($\wedge$, $\vee$, $\neg$,
entity membership, etc.). From these losses, one can argue where to tweak if sanctity is not maintained.

[Meta-Review · NeurIPS 2019]

The reviewers have different views on this paper - although the experimental results are not very strong the paper is well-written and introduces some interesting new ideas to the NeurIPS community. Overall I think the paper is worth presenting at NeurIPS. However, the final camera-ready paper MUST discuss the relation to heirarchical embedding schemes such as [1,2], as discussed by (R3) and also logic tensor networks [3], a related formalism for embedding logical expressions. [1] Poincaré Embeddings for Learning Hierarchical Representations. Maximilian Nickel, Douwe Kiela. NIPS, 2017. [2] Learning Continuous Hierarchies in the Lorentz Model of Hyperbolic Geometry. Maximilian Nickel, Douwe Kiela. ICML, 2018. [3] Serafini, Luciano, and Artur d'Avila Garcez. "Logic tensor networks: Deep learning and logical reasoning from data and knowledge." arXiv preprint arXiv:1606.04422 (2016).